# A Smart Service Platform for Cost Efficient Cardiac Health Monitoring

**DOI:** 10.3390/ijerph17176313

**Published:** 2020-08-30

**Authors:** Oliver Faust, Ningrong Lei, Eng Chew, Edward J. Ciaccio, U Rajendra Acharya

**Affiliations:** 1Department of Engineering and Mathematics, Sheffield Hallam University, Sheffield S1 1WB, UK; N.Lei@shu.ac.uk; 2Faculty of Information Technology, University of Technology Sydney, Ultimo, NSW 2007, Australia; Eng.Chew@uts.edu.au; 3Department of Medicine—Cardiology, Columbia University, New York, NY 10027, USA; edwardciaccio@gmail.com; 4Biomedical Engineering Department, Ngee Ann Polytechnic, Singapore 599489, Singapore; aru@np.edu.sg; 5Department of Bioinformatics and Medical Engineering, Asia University, Taichung 41354, Taiwan; 6School of Management and Enterprise, University of Southern Queensland, Springfield, QLD 4350, Australia

**Keywords:** service platform, internet of things, e-health, deep learning, heart rate

## Abstract

Aim: In this study we have investigated the problem of cost effective wireless heart health monitoring from a service design perspective. Subject and Methods: There is a great medical and economic need to support the diagnosis of a wide range of debilitating and indeed fatal non-communicable diseases, like Cardiovascular Disease (CVD), Atrial Fibrillation (AF), diabetes, and sleep disorders. To address this need, we put forward the idea that the combination of Heart Rate (HR) measurements, Internet of Things (IoT), and advanced Artificial Intelligence (AI), forms a Heart Health Monitoring Service Platform (HHMSP). This service platform can be used for multi-disease monitoring, where a distinct service meets the needs of patients having a specific disease. The service functionality is realized by combining common and distinct modules. This forms the technological basis which facilitates a hybrid diagnosis process where machines and practitioners work cooperatively to improve outcomes for patients. Results: Human checks and balances on independent machine decisions maintain safety and reliability of the diagnosis. Cost efficiency comes from efficient signal processing and replacing manual analysis with AI based machine classification. To show the practicality of the proposed service platform, we have implemented an AF monitoring service. Conclusion: Having common modules allows us to harvest the economies of scale. That is an advantage, because the fixed cost for the infrastructure is shared among a large group of customers. Distinct modules define which AI models are used and how the communication with practitioners, caregivers and patients is handled. That makes the proposed HHMSP agile enough to address safety, reliability and functionality needs from healthcare providers.

## 1. Introduction

Heart Rate Variability (HRV) is a good indicator of human health, which can be used to detect Atrial Fibrillation (AF), sleep disorders, Cardiovascular Disease (CVD), and diabetes. These non-communicable diseases cause major public health problems. The Background section provides some context on these diseases by reviewing relevant scientific work. HRV can be used for cost efficient and unobtrusive disease diagnosis and treatment monitoring. Therefore, this technology has the potential to play a major role in systems which address these public health problems. In order to realize this potential, a framework is needed which turns the versatility of HRV into tangible benefits for patients and public health providers. Heart Rate (HR) signals report the beat-to-beat interval of the human heart. Due to natural HR fluctuations, the data rate varies from about 2 bytes to 8 bytes a second. The high information content and low data rate makes HR data ideal for cost effective e-health applications. To be specific, the data can travel from the patient over most of the wired and wireless infrastructure to cloud services based on Internet of Things (IoT) protocols. The ability to move HR data from the patient to a central and universally accessible location has the advantage that the disease detection can be independently verified by human practitioners, and the data can be used to improve the decision-making processes which ultimately improves the diagnostic quality. Another advantage of the low data rate and the associated low processing requirements is real-time disease symptom detection. This is important, because disease symptoms can be intermittent and, during an acute phase, a particular risk may significantly increase. For example, the stroke risk in paroxysmal (intermittent) AF has a five-fold increase when the AF is active [1]. Hence, it is important to monitor the disease and act immediately to protect the patient’s life. The real-time requirement poses a significant challenge to the technology which underpins a health monitoring service. Apart from these technological challenges, the time pressure during real-time monitoring raises also safety concerns for the diagnosis process. To be specific, evidence which underpins the diagnosis must be preserved in order to conduct a root cause analysis in case an undesired outcome has occurred. To maintain the evidence is also important for future training and testing purposes.

Rapidly evolving information technologies underpin the HR data acquisition, communication, and storage processes. A modular service platform can help to translate data-centric information into knowledge, which improves diagnosis and treatment monitoring. Hence, this approach can enhance the quality of healthcare and reduce the cost. A service platform is defined as a specific set of service functionality that is used across multiple services, or the procedural connections that bridge and link specific sets of service functionality [2]. The platform functionality is realized with a modular service architecture. The architecture is based on the concept of common and distinct modules [3]. The common modules are shared among multiple services. This resource sharing leads to cost savings. In contrast, distinct modules implement the functionality to address specific customer needs. The challenge for the platform design is to find a balance between commonality and specificity, such that the resulting services are safe, reliable, and functional in a cost-effective way.

With this paper, we propose a smart Heart Health Monitoring Service Platform (HHMSP) based on HR signals. We address the need for a cost effective monitoring and diagnostic process that can be used for a wide range of non-communicable diseases, including AF, CVD, sleep disorders, and diabetes, in a safe and reliable way. The platform was created by combining readily available technology. The signal measurement was done with commercial HR sensors. The acquired HR data was relayed via smart phones to an IoT framework. The framework provides almost universal access to the data. Having access to the data anytime anywhere allows us to interpret the signals through machine classification or human expertise. Furthermore, the stored data can be used for root cause analysis and future improvements to the analysis process. Figure 1 shows an overview diagram of the proposed service platform. The cloud server, at the center, incorporates the machine classification functionality. We envision that, for routine disease monitoring tasks, the data interpretation is done by machine classification, and human experts just validate the machine decisions. This hybrid approach, where machines and humans work together cooperatively, improves cost efficiency while maintaining the safety and reliability of the diagnosis. With respect to cost, the HHMSP harvests the economies of scale with common modules that implement an IoT infrastructure. These modules handle the data communication from the point of measurement to a central storage location. Distinct modules facilitate both decision-making processes and the resultant dissemination. The concept of common and distinct modules makes the HHMSP sufficiently agile to cope with future requirements.

To support our claim that the proposed service platform is safe, reliable, and functional, as well as cost effective, we have structured the remainder of the manuscript as follows. The Background section introduces the public health problems caused by CVD, AF, diabetes, and sleep disorders. Furthermore, we show how disease symptoms can be detected in HR signals. The section details the service platform architecture. Subsequently, the merits and indeed the limitations of this architecture are discussed. The Conclusion wraps up the paper and summarizes the main points.

## 2. Background

The medical justification for the proposed HHMSP comes from the fact that HR signals are a good indicator of human health [4]. To be specific, the beat-to-beat intervals of the human heart hold valuable information about the Autonomic Nervous System (ANS). The information content of HR is similar to the information content of other physiological signals which have higher requirements for measurement setup as well as communication and processing infrastructure [5]. For example, Electrocardiogram (ECG) signals record more details of the human heart’s electrical activity when compared to HR [6]. Hence, the ECG measurement setup is more complex, and more resources are required to communicate and process the signal. As a result, the operating cost of diagnostic support processes that use ECG is higher when compared to those that use HR.

The information contained in HR signals can be used to monitor AF, CVD, sleep disorders, and diabetes. Formalizing the information extraction is a major challenge for practical monitoring systems. Most of these systems depend on feature engineering and machine learning. The next sections provide some background on the economic cost of these diseases and they outline how HR signals can support diagnosis and treatment monitoring.

### 2.1. Atrial Fibrillation

Worldwide, about 33.5 million suffer from AF, which makes it the most prevalent sustained rhythm disorder [7]. In developed countries, the majority of AF patients are aged 65 years and above. There are two types of AF, namely persistent and paroxysmal (intermittent) AF. Both types are associated with adverse health effects, through thromboembolic complications, heart failure, and stroke. During an active period, AF increases the stroke risk five-fold [1]. The combination of all these negative consequence makes AF an independent predictor for death. Untreated AF poses a significant cost burden on healthcare systems worldwide, because of increased morbidity- and mortality-associated therapeutic interventions [8]. Public Health England estimated that only about 79% of all AF cases are detected. Based on this estimate, they put forward that optimizing risk factor detection and subsequent decease treatment will return £2.30 for every £1 spent (https://www.gov.uk/government/publications/health-matters-preventing-cardiovascular-disease/health-matters-preventing-cardiovascular-disease).

AF changes the beat to beat interval of the human heart [9]. However, it is difficult and time consuming for a cardiologist to detect these changes [6]. Therefore, computer support is essential for HR based AF diagnosis. Table 1 provides an overview of HR based AF detection since 2016. As such, the detection process involves either classical machine learning or Deep Learning (DL). Classical machine learning requires feature engineering, basically extracting the information from the signal first before making a decision. DL does not require explicit feature engineering. This has the advantage that the decision-making algorithm sees the complete signal, and hence a decision is based on all of the available information. Studies have shown that DL algorithms outperform classical machine learning, based on feature engineering, for larger data sets, i.e., big data [10]. That is the reason why the more recent studies employ this type of decision-making. All studies listed in Table 1 were based on data from the publicly accessible PhysioNet database [11]. That creates a competitive environment with a clear history of improvement. To be specific, it is possible to compare the study results with a fair bit of understanding about common and distinctive aspects of the work.

### 2.2. Sleep Disorders

Sleep is essential for physical and mental health. It is an active and regulated process that metabolizes energy, matures neural connections, and consolidates learning as well as memory [19]. Unfortunately, a faster paced rhythm of life and other detrimental lifestyle changes cause a higher prevalence of sleep disorders [20]. These disorders cause daytime sleepiness which has a negative impact on public safety and reduces the quality of life [21,22]. Epidemiologic studies show that up to 24% of the adult population has some sort of sleep disorder [23,24]. Wickwire et al. found that the annual cost for the global economy caused by sleep disorders exceeds 100 billion USD per year [25]. Ozminkowski et al. estimated that, within a six month period, the average direct and indirect costs for adults with sleep disorders were about 1000 USD greater than for normal control subjects [26].

The ANS activity reflects individual sleep stages [27]. During Rapid Eye Movement (REM) sleep, the respiratory rate is irregular and the overall lung tidal volume decreases when compared to Non-Rapid Eye Movement (NREM) sleep [28]. Consequently, both HR and respiratory rate change according to the sleep stages [29]. HRV measures can be used to detect changes in HR signals. These measures may provide useful information about the ANS state, which is relevant for a medical diagnosis. During REM sleep, the ANS fluctuations between sympathetic and parasympathetic activities. That increases the statistical measures of mean HR and HR variability. Therefore, it is possible to find HRV features which discriminate between NREM and REM sleep. The feature extraction process is based on statistical, time-, frequency-domain, and nonlinear analyses algorithms. Trinder et al. used HRV measures to analyze the autonomic activity during sleep [30]. Penzel et al. extracted spectral analysis and detrended fluctuation features to gain sleep stage information [31]. Table 2 summarizes scientific work that used HR signals to detect sleep disorders. The sparsely populated result columns indicate that HR for sleep disorder detection is a research area which is still in its initial exploration phase. Scientific studies, like the most recent one by Yajima et al. [32], are still concerned with feature engineering, which aims to establish the role of HR based sleep disorder detection. The studies, detailed in Table 2, were based on private data sets. That makes it difficult to compare and indeed to repeat the studies.

### 2.3. Cardiovascular Disease

CVD describes a class of diseases, including, but not limited to, Coronary Artery Disease (CArD) and Congestive Heart Failure (CHF). Worldwide an estimated 422.7 million patients suffer from CVD [36]. 49 million people live with this disease in the EU. In 2016, the World Health Organization (WHO) estimated that 17.9 million people died from CVD globally (www.who.int/news-room/fact-sheets/detail/cardiovascular-diseases-(cvds)). This represented 31% of global death. It is estimated that CVD costs the EU economies €210 billion (≈240 USD) a year [37]. CVDs are more prevalent with age, due to prolonged accumulation of plaque and vascular aging [38,39]. Other more controllable risk factors for CVD are e.g., high blood pressure, high cholesterol levels, obesity, diabetes mellitus, and stress [40,41].

CArD describes a condition where the coronary arteries cannot deliver enough oxygen and nutrients to the heart muscle [42]. That reduced blood flow is caused by a decreased artery diameter. CArD is one of the root causes for Sudden Cardiac Death (SCD) [43]. Table 3 summarizes scientific work that used HR signals to detect CArD. With the advent of DL the research field began to mature. Lih et al. did not report the best accuracy in Table 3, but their study involved the detection of CArD, Myocardial infarction, and CHF [44]. Posing such a multi-class problem indicates a high degree of data handling sophistication which might be sufficiently robust to support clinical practice. Table 3 shows that public data sets are available which can be used for CArD detection. That provides some means to compare and repeat the studies.

CHF causes high morbidity and mortality. Worldwide, around 26 million people suffer from this condition [50]. Long term monitoring is needed to collect sufficient evidence for a reliable diagnosis of this cardiovascular disease. Binder et al. used HR signals to derive a prognosis for CHF patients [51]. CHF changes the heart rhythm. Studies that employ feature engineering and threshold techniques, such as Hsuet al. [52] and Liu et al. still aimed to establish that these changes are reflected in HR traces [53]. Whereas DL approaches aim to automate the detection process and thereby improve clinical workflows. Table 4 summarizes scientific work that detected CHF in HR signals. CHF detection systems were designed based on publicly available databases. Hence, it is possible to repeat and compare the study results.

### 2.4. Diabetes

The WHO estimates that there were 422 million people living with diabetes in 2014 (www.who.int/news-room/fact-sheets/detail/diabetes). The disease is associated with long-term complications that result from damaged blood vessels. As such, diabetes is not a deadly disease, however recent studies indicate that having diabetes doubles the risk of death [58]. Bommer et al. estimated that the global economic burden will rise from 1.3 trillion USD in 2015 to 2.5 trillion USD in 2030 [59]. T. M. Dall put forward that undiagnosed and poorly managed diabetes has a high economic burden on the US. Hence, finding cost-effective methods to detect and monitor this disease is of high importance [60].

Currently, diabetes is diagnosed based on blood tests [61]. Taking blood from patients is invasive and there are non-trivial risks associated with this action (https://www.medic8.com/blood-disorders/blood-test/risks-of-blood-tests.html). HRV analysis can provide a non-invasive alternative to blood testing for initial diagnosis [62]. The reason why diabetes can be detected in this manner comes from the fact that the disease alters the ANS and these alterations impact on the HR [63,64]. Acharya et al. used HRV to establish an integrated diabetes index which can help clinicians during the initial diagnosis [65]. Swapna et al. used higher order spectra to detect diabetes based on HR signals [66]. The non-invasive nature, coupled with the low data rate and the simple measurement setup, make HR signals ideal for treatment monitoring. Table 5 summarizes scientific work on automated detection of diabetes based on HR signals. All studies on HR based diabetes detection were based on private data sets. Hence, it is difficult to repeat and compare the study results.

### 2.5. Research Gap

The literature review highlights the economic cost of AF, sleep disorders, CVD, and diabetes. Furthermore, it establishes that computer based algorithms can be used to detect disease symptoms in HR signals. However, we found that there is no agreement on which features work best for a given problem. Furthermore, state-of-the-art features are used for both human and machine decision-making. Having similar features for both decision processes raises safety concerns, because the decisions are not independent, i.e., they might be based on the same erroneous feature. That may be the reason why the published studies fail to describe a system which can be used to make the theoretical results work for patients in a real-world scenario. Related to the lack of system proposals, a business strategy on how to build cost-effective, scalable, and safe diagnostic systems is also missing. In the next section we tackle these problems by introducing a service platform architecture which addresses both patient and economic needs in an economical way. Implementing this architecture creates a multi-disease monitoring platform which is both practical and cost-effective.

## 3. System Architecture

The HHMSP allows us to frame the health monitoring problem in a business environment which demands cost-effective solutions. Furthermore, the business centric framing mandates that the healthcare provider becomes a customer who uses a service to improve disease diagnosis or disease monitoring. The healthcare provider registers a patient with a service, which is tailored for an individual disease. For example, a cardiologist registers a patient with the AF monitoring service. That service will provide diagnosis and treatment support.

The HHMSP architecture was designed such that it incorporates HR measurement, IoT for data distribution, cloud servers for storage and advanced Artificial Intelligence (AI) for data interpretation. The block diagram, shown in Figure 2, provides a graphical representation of the proposed HHMSP architecture.

Implementing the platform architecture establishes the service functionality, which can be described as follows: The healthcare provider purchases a specific diagnosis support service (The specific service offering will depend on the service provider business model and on regulatory requirements.). At the clinic, a patient is fitted with a HR sensor and registered with the service platform (DB module). The patient wears the sensor which communicates the measurement data to a data acquisition module. That module relays that data to an IoT cloud server. Once on the cloud server, the HR data can be used for diagnosis support. The deep learning module queries the data from the cloud and extracts relevant information with a disease specific model. The alarm module informs the physician if the DL results meet a specific condition, for example a disease is detected. Once alerted, the medical expert can use the physician support module to interact with both the DL result and raw HR data during the diagnostic process. Having two cascaded decision processes, where a DL algorithm provides an initial assessment and a human practitioner reaches a diagnosis, improves the safety when compared to each individual decision system. The feedback module can be used to disseminate rapid response warnings when an immediate health risk was detected, and it can be used for patient compliance, i.e., a request to wear the sensor. A diagnosis should be communicated via the standard channels established by the healthcare provider. The following sections describe both common and distinct modules of the proposed HHMSP. To illustrate the platform functionality, we discuss the details and state performance results from an AF monitoring service.

### 3.1. DB Module

The DB module incorporates a Database (DB) which holds patient information. Necessary fields are: Patient ID, disease type, assigned physician, cloud server login key, service start date, service end date. For the HHMSP, the disease type can be sleep disorders, CVD, diabetes or AF. Optional fields are: Patient name and detailed feedback. The feedback could be a list of healthcare instructions for the patient. At the start of the service subscription, the caregiver registers the patient with the DB module. In response, the DB module will provide the cloud server login key. That login key is used to set up the data acquisition module, as described in the next section.

### 3.2. Data Acquisition Module

The measurement setup is straight forward. The patient wears a breast strap with an embedded HR sensor. The sensor picks up the electrical activity which indicates the human heartbeat, known as the R peak. The time between two consecutive R peaks is measured and the resulting value is communicated to the mobile phone as the RR interval. A stream of RR intervals is referred to as HR signal. We tested our system with two commercially available sensors, namely Polar H10 and Polar H7 (www.polar.com/uk-en). Low power Bluetooth is used to transfer the data from the HR sensor to an Android smartphone or tablet.

The heart health app relays the sensor information to the IoT cloud server. The data exchange with the IoT cloud server is facilitated by an Internet connection. Figure 3 shows a screen shot of the heart health app, which was created by extending the BLEConnect program (https://github.com/mobilars/BLEConnect). The background shows an averaged HR trace. The dialogue in the foreground requests the user to enter the login data for the IoT server. Each user has a unique cloud server login key. Both the patient and authorized caregivers can access the patient’s data using the same cloud server login key, which is stored in the DB module.

### 3.3. Cloud Server

We have used ThingSpeak from Mathworks (https://thingspeak.com/) as cloud server. To test the HHMSP, two accounts were created with test data from PhysioNet’s AFDB and two accounts were set up with measured data. Each account has two channels. The first channel, which is called RR_interval_data, holds the measurement data. The content is updated when the heart health app relays the HR signals. The second channel, which is called AF_detection_result, holds the machine classification results. The result channel content is updated when the machine classification system produces a new result.

### 3.4. Deep Learning Module

The deep learning module reads new entries from the RR_interval_data channel and writes the classification outcomes to the AF_detection_result channel. The DL system was used because of its ability to detect the disease based on hidden information present in the HR signals. As such, DL does not require feature engineering. This is a significant improvement compared to conventional machine learning algorithms [10]. The absence of feature engineering makes DL robust when it classifies samples from a larger data set. In a clinical setting, this robustness translates to high-quality diagnostic support for patients using HR signals. Each disease requires a distinct DL model. Based on information from the DB module (disease type), the deep learning module selects the appropriate model for the HR to be analyzed.

The AF module was created by training the DL system using labeled HR data obtained from PhysioNet’s AFDB (www.physionet.org/physiobank/database/afdb/). Based on data from 20 subjects, the system achieved an accuracy of 98.51% with 10-fold cross-validation [15]. The blindfold classification performance, with 99.77% accuracy, indicates that the model is able to deliver high quality diagnostic support for patients using HR signals.

### 3.5. Alert Module

The alert module prompts the physician to use the assigned physician support module. That prompt is triggered when a specific condition is reached, usually when a disease is detected. Sending the alert message can be facilitated with a range of communication channels, such as e-mail, Twitter, and instant messaging.

For the AF monitoring service, the alert module is implemented on ThingSpeak. A monitor program checks every DL result on the AF_detection_result channel. If the estimated AF probability is above 0.5, the monitor program will send an alert message to the assigned physician via Twitter and SMS.

### 3.6. Physician Support Module

The assigned physician has access to the HR data and DL diagnostic support through the HHMSP. As such, HR signals are complex and their waveform resembles noise. Therefore, disease detection, with the naked eye, is inefficient and error prone. Algorithm support is needed to extract relevant information from the signal. The HHMSP physician support module incorporates algorithm support in the form of DL results and features. The use case scenario for the module unfolds as follows. The assigned physician receives an alert from the alert module with information about the patient condition. Figure 1 shows an example where the physician received an alert message, because the DL system detected an AF episode in the HR signal from one of the patients. By using the physician support module, the assigned physician confirms or rejects the DL assessment and establishes a diagnosis [72]. Then the diagnosis will be communicated to the patient and other stakeholders via the feedback module, as discussed in the next section.

Figure 4 shows a screenshot of the prototype HHMSP physician support module with the AF GUI. This module was developed by extending the Heart Rate Variability Analysis Software (HRVAS) (https://github.com/jramshur/HRVAS). The drop-down list allows the assigned physician to select a patient based on the patient ID, as provided by the DB module. The left-hand side of the figure visualizes the automated decision support from the deep learning module. The right-hand side shows the HRV signal features which were designed to help the assigned physician to validate the DL results.

In the example, shown in Figure 4, the patient with ID 08455 was selected, because of an alert message. After pressing ”Fetch Data” the DL results are displayed as estimated AF probability over time, in the upper left corner of the figure. In the timeline, the assigned physician can select a region of interest. Once the region is identified, the support tool fetches the corresponding HR signal segment and calculates a wide range of features from that segment. The feature extraction results are displayed on the right side. Figure 4 shows the selected region of interest as a light red square in the timeline plot. The square area encompasses a spike in estimated AF probability and a step edge. The estimated AF probability is reflected in the color scheme of the HR graph shown in the lower left corner. The color bar maps the estimated AF probability to a specific color, for example a low probability is blue and a high probability is red. The small red region in the HR graph reflects the spike. Similarly, the red region on the right of the HR graph reflects the step edge. The Poincaré plot [73], as well as features SD1 and SD2, for the selected HR signal segment are displayed on the right side of Figure 4.

### 3.7. Feedback Module

Once the assigned physician has reached a diagnosis, the feedback module is used to communicate with the patient. Social media, e-mail and personal phone calls can be used to provide feedback. One way of providing feedback is with a simple traffic light system: Green—all is well. Yellow—take predetermined precautionary action. Red—see your physician immediately.

### 3.8. Testing

The HHMSP was tested with the AF monitoring service. By doing that, we tested all common modules for data handling and the specific modules which establish diagnostic support. To verify all aspects of the AF monitoring service, we registered two healthy volunteers with the DB module. Subsequently the volunteers were fitted with HR sensors and their data was communicated and processed by the HHMSP. To be specific, their data was processed with the deep learning module which executed the AF specific DL model. However, the DL results were not verified by a cardiologist. Therefore, no diagnosis was established and the results have only an indicative character.

To validate both deep learning and physician support modules for the AF monitoring service, we streamed benchmark data from PhysioNet’s MIT-BIH Atrial Fibrillation Database [74] to the data acquisition module. Figure 4 shows a screenshot of the physician support module which details both DL and feature extraction results for the benchmark data. Apart from visual inspection, enabled by the physician support module, we also compared the deep learning module results for the benchmark data with the results for the same benchmark data obtained by a previous study on the automated detection of atrial fibrillation using a long short-term memory network with RR interval signals [15]. In that study, we used the same model as in the deep learning module. We established that the DL results were the same. With that, we could validate both data acquisition and deep learning modules. Furthermore, that validation confirmed that the AF monitoring service achieved 99.77% accuracy with benchmark data.

## 4. Discussion

In this paper, we propose a practical and innovative application of IoT networks to tackle public health problems caused by non-communicable diseases with a cost-effective service platform. Having such a service platform leads to improved diagnosis and treatment monitoring processes. Currently, CVD, AF, diabetes, and sleep disorders have distinct diagnosis and treatment monitoring processes. Diabetes and sleep disorders are diagnosed based on measurements taken in a clinical environment. Thus a patient must travel to the clinic whenever measurements are necessary. This process is inconvenient for the patient and the cost is high. Furthermore, there is the possibility that patients are symptom-free during the measurement. AF diagnosis is based on Holter monitoring [75] which is a process to capture ECG signals in the patient environment. To start the process, a patient has to travel to a medical facility where the Holter monitor fitted. After the measurement duration, usually 72 h, the patient has to travel again to the medical facility to return the Holter monitor. The data analysis can only start after the Holter monitor is returned, which delays the diagnosis. Furthermore, a 72 h measurement duration might be insufficient to detect an AF period if the patient suffers from the paroxysmal type of the disease. An AF monitoring service, based on the proposed service platform, requires the patient to travel to the clinic only once for the initial fitting of the HR sensor. After that, the HR data travels rather than the patient. Reduced patient traveling and prolonged measurement duration are distinct advantages of the HHMSP based diagnostic and treatment monitoring process. Less traveling makes the process more convenient for the patient. The prolonged measurement duration means that more cases of a specific disease are detected, which improves patient outcome. However, prerequisite for such a change in diagnosis and treatment monitoring processes are safety considerations.

Our approach is safe, because we use a cascaded decision-making process. The service platform is also practical, because state-of-the-art IoT technology is able to communicate, store and process HR data in a cost-effective way. The service platform can be integrated into existing e-health solutions, such as Electronic Patient Record (EPRs). The innovation, which sparked the idea for the HHMSP, is DL. To be specific, DL is able to differentiate disease and non-disease classes accurately and in real time. Indeed, DL makes the proposed multi-disease monitoring service platform feasible. Traditional Computer-Aided Diagnosis (CAD) systems require feature engineering, which restricts the information available to the machine learning algorithm. This is the reason why machine learning fails for big data. Furthermore, a service platform, based on classical machine learning, would be more complex, because each service would require a feature engineering module and a classification module. Neither feature engineering nor classification modules are uniform. For example, the number and type of features as well as machine learning algorithm for AF detection might be quite different from the algorithms used for diabetes detection. In contrast, the architecture for the proposed service platform is uniform—only the DL model changes, but the algorithm which executes that module stays the same. The HR data is stored in the cloud server module, which allows real-time universal access to the data. That access is important during the two-stage diagnostic process, which combines machine and human decision-making. The two-stage diagnostic process offers three main advantages: (1) Safety through human checks and balances, (2) significantly reduced physician workload, and (3) increased efficiency, which enables real-time diagnosis.

In order to apply the service platform concept, we had to frame the public health problems in terms of a market driven supply and demand scenario. Healthcare providers become customers who have specific needs. These needs are identified through market analysis and subsequent market segmentation. As it stands, each disease is a market segment and the needs of that market segment are addressed by a specific health monitoring service. In terms of architecture, an individual service is realized by combining common with distinct modules. The modular HHMSP has the capability to extend the service offerings by adding new distinct modules for a newly identified market segment. For example, the AF monitoring service uses all the common modules, together with the distinct AF Model and AF GUI.

The fact that DL does not require feature engineering is significant, because that makes it independent from human decision-making. In contrast, classical machine learning requires feature engineering [73,76] and therefore it is not independent from human decision-making. From a human perspective, HR signals are noise-like, and it is difficult, even for a skilled practitioner, to recognize the underlying structure of the signal. Hence, human HR interpretation must be supported by features. These features are similar to the features used for machine classification. Therefore, a mistake in the feature engineering is likely to affect both classical machine learning and human decision-making. From a safety perspective, having both decision systems depend on the feature extraction methods constitutes a single point of failure. That means an error in the feature engineering might lead to the same erroneous conclusion from a machine learning algorithm and a human practitioner. Such an erroneous conclusion might harm patients, making the system less safe. In contrast, DL does not require explicit feature engineering, that makes a DL result independent from any decision that was based on feature engineering coupled with a decision-making process. Hence, there is a smaller probability that errors overlap and both the human and DL algorithm come to the same erroneous conclusion. Therefore, the combination of DL and human decision-making is safer when compared to feature based machine learning and human decision-making.

### 4.1. Limitations

HR has a low data rate and a high information content. That is beneficial for data communication, storage and processing. However, having high information content implies that the waveform is noise-like. Hence, it is difficult for humans to interpret the signal. Unlike ECG based diagnosis, it is difficult to predict a specific disease from the time domain HR waveform. Therefore, computer support is needed to interpret HR signals. That computer support, for physician decision-making, comes in the form of features. It takes trust and effort on the part of the medical practitioner to work with these features.

### 4.2. Future Work

The service platform concept offers the flexibility needed to address diverse customer needs. Individual diseases require customized services to address the different needs for diagnosis and treatment monitoring. The generative innovation opportunities offered by the platform is potentially significant. For example, we could link the HHMSP with existing Hospital Information Systems (HISs) to enhance patience experience and improve outcomes.

The HHMSP was tested with benchmark data rather than with a medical expert who establishes a diagnosis. Future work needs to address this shortcoming. To be specific, we propose to validate the AF monitoring service with HR data measured from patients. In order to involve patients and medical experts in the next validation step, we have to address shortcomings in the way the sensor data is handled. To be specific, currently only standard protocols were used for the Data acquisition module. These protocols follow no or only weak data security standards. Hence, the data is at risk of being captured with negative consequences for the patient. This needs to be addressed with end-to-end encryption from the sensor to the cloud server. Furthermore, for clinical studies involving patients, the sensor and data handling technology should conform to health, safety, and environmental protection standards. The sensors used to test the platform prototype fall short of the health protection standard required for medical devices.

From the business perspective, the HHMSP customers, i.e., the healthcare providers, offer professional support to the platform to ensure the services are consistent with the medical standards of the country in which the heart health monitoring service is provided. In the future, we need to address the mutual accountabilities of the platform owner and the healthcare provider. Having a clear process which assigns accountability is a prerequisite to understand and subsequently improve the systemic safety.

## 5. Conclusions

The proposed HHMSP is based on HR signals. There are two reasons for this type of design. First and foremost, HR signals hold diagnostically relevant information about a wide range of diseases, including, but not limited to CVD, sleep disorders, AF, and diabetes. The non-invasive nature and easy measurement setup makes HR ideal for patient led data acquisition. The second factor is the low data rate. The low data rate allows us to use low energy Bluetooth to connect the sensor, which extends the battery life and/or enables the sensor to be smaller. Another positive aspect is that data communication, storage, and processing is cheap. However, data interpretation is complex. Physicians require computer support to make sense of the data. Even with computer support, it is impossible for a physician to monitor the data in real time, because of the streaming nature of the data. Therefore, we put forward DL for real-time data processing. Physicians are only involved to check the machine classification results. These human checks and balances add another layer of safety, i.e., the diagnosis is only partially automated, and the responsibility still rests with the human decision maker.

The proposed HHMSP architecture balances the commonality and distinctiveness, such that it is possible to address a wide range of customer needs, and at the same time harvest the economies of scale. The common modules establish the IoT functionality which is responsible for communicating and storing the HR data. Distinct modules facilitate multi-disease diagnostic support through machine classification and result dissemination. With the proposed architecture, we address the need of patients with a specific disease through distinct modules, while at the same time harvesting the economies of scale with common modules. 

## Figures and Tables

**Figure 1 ijerph-17-06313-f001:**
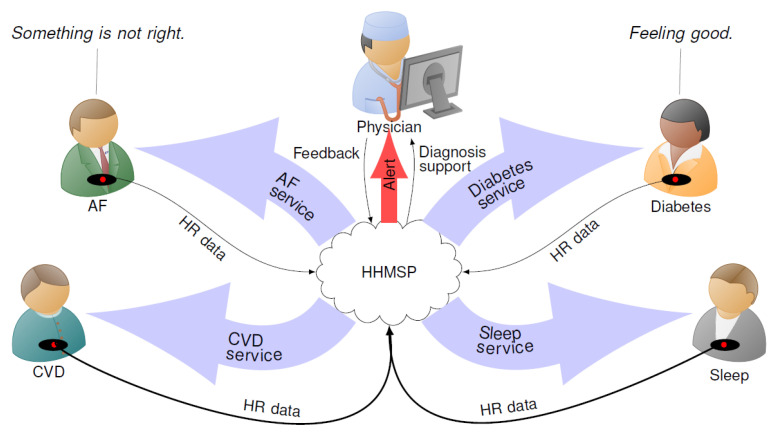
Overview diagram of the Heart Health Monitoring Service Platform (HHMSP).

**Figure 2 ijerph-17-06313-f002:**
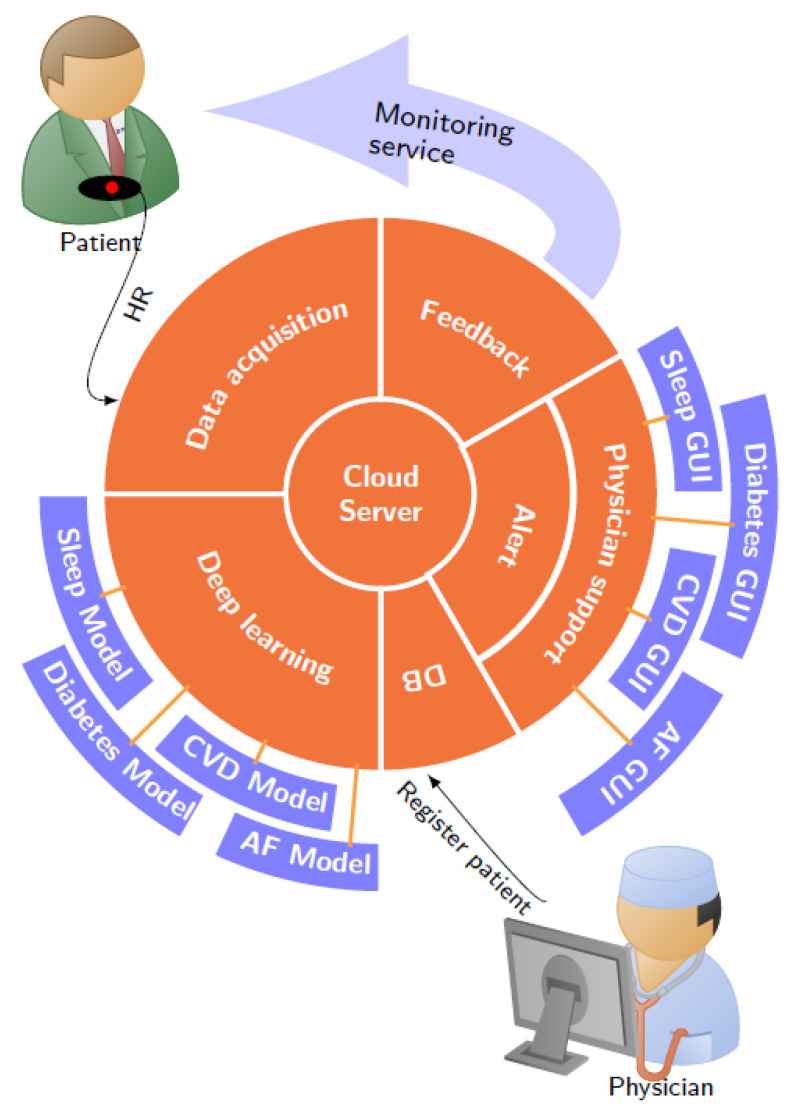
Heart Health Monitoring Service Platform (HHMSP) architecture. The common modules are in orange and the distinct modules are in blue. Abbreviations in the figure: Cardiovascular Disease (CVD), Atrial Fibrillation (AF), and Graphical User Interface (GUI).

**Figure 3 ijerph-17-06313-f003:**
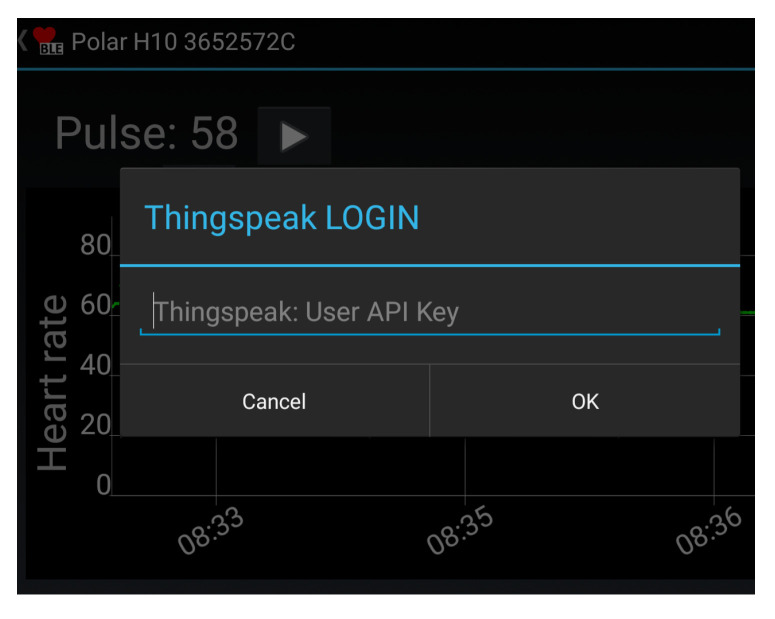
Screen shot of the heart health app which relays the sensor data to the cloud server.

**Figure 4 ijerph-17-06313-f004:**
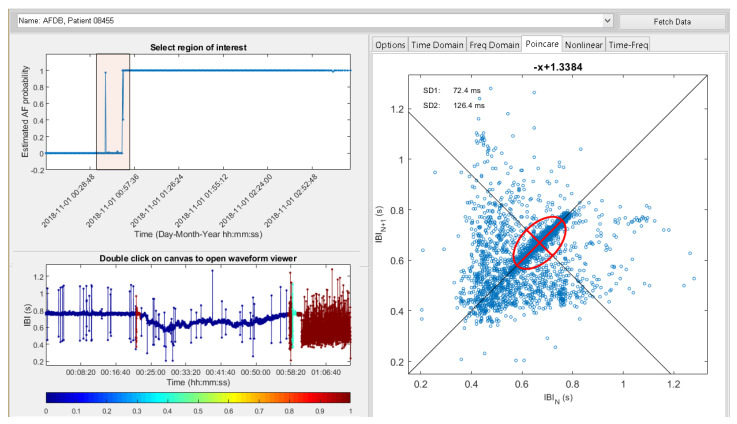
Physician support with the modified Heart Rate Variability Analysis Software (HRVAS) program.

**Table 1 ijerph-17-06313-t001:** Methods for Heart Rate (HR) based detection of Atrial Fibrillation (AF). Abbreviations in the table: Acc, Sen, Spe, Deep Learning (DL), and Normal Sinus Rhythm (NSR).

Author	Data and Method	Performance		
		Acc/%	Sen/%	Spe/%
Faust et al. 2020 [12]	Physionet Long Term AF Database. DL with hold out validation	94	-	-
Ivanovic et al. 2019 [13]	Private data set. DL	89.67	94.20	-
Andersen et al. 2019 [14]	Physionet: MIT-BIH AF Database, MIT-BIH Arrhythmia Database, and MIT-BIH NSR Database DL	-	98.98	96.95
Faust et al. 2018 [15]	MIT-BIH Atrial Fibrillation Database. DL	98.51	98.32	98.67
Henzel et al. 2017 [16]	MIT-BIH Atrial Fibrillation Database. Linear measures & Threshold	93	90	95
Cui et al. 2017 [17]	MIT-BIH Atrial Fibrillation Database Ensemble model & Threshold	97.78	97.04	97.96
Islam et al. 2016 [18]	MIT-BIH AF database and MIT-BIH Arrhythmia database. Entropy & Threshold	96.38	96.39	96.38

**Table 2 ijerph-17-06313-t002:** Methods for HR based detection of sleep disorders.

Author	Data and Method	Performance		
		Acc/%	Sen/%	Spe/%
Yajima et al. 2019 [32]	One case: 52-year-old man. Statistical features and statistical analysis	-	-	-
1-5 Tripathy et al. 2018 [33]	Fourteen subjects (six female and eight male). Recurrence quantification analysis and dispersion & DL	95.71%	-	-
1-5 Yoon et al. 2017 [34]	Twenty-one healthy subjects (male: 12, female: 9) and 30 subjects (male: 25, female: 5) with obstructive sleep apnea (OSA). Statistical parameters, Spectral power, variability measurements. Threshold decision	87.54%	-	-
1-5 Liu et al. 2017 [35]	Seventy-five sleep apnea patients. Time domain statistical parameters, Spectral power, nonlinear measurements evaluated with statistical methods	-	-	-

**Table 3 ijerph-17-06313-t003:** Methods for HR based detection of Coronary Artery Disease (CArD).

Author	Data and Method	Performance		
		Acc/%	Sen/%	Spe/%
Lih et al. 2020 [44]	Physionet data, no further specifications. DL	98.5%	99.30%	97.89%
Shi et al. 2019 [45]	Fantasia and St. Petersburg Institute of Cardiological Technics 12-lead Arrhythmia databases. Wavelet and entropy & K-Nearest Neighbour (K-NN)	97.5	100	95
Venkatesh et al. 2018 [46]	Privaate data set. Statistical analysis	-	-	-
Singh et al. 2018 [47]	NSR Physionet and St. Petersburg Institute of Cardiological Technics database. Generalized discriminant analysis & extreme learning machine	100	-	-
Yıldız et al. 2016 [48]	Private data set. Statistical analysis	-	-	-
Kumar et al. 2016 [49]	Private data set. FAWT and entropy & LS-Support Vector Machine (SVM)	100	-	-

**Table 4 ijerph-17-06313-t004:** Methods for HR based detection of Congestive Heart Failure (CHF).

Author	Data and Method	Performance		
		Acc/%	Sen/%	Spe/%
Hsuet al. 2019 [52]	BIDMC Congestive Heart Failure Database, MIT-BIH Normal Sinus Rhythm Database, and Long Term AF Database. Average entropy & Threshold	90	-	-
Liu et al. 2019 [53]	Physionet CHF database. Statistical measures & analysis	-	-	-
Wanget et al. 2018 [54]	Physionet CHF database. DL	85.13	-	-
Yoon et al. 2017 [55]	MIT-BIH Atrial fibrillation, MIT-BIH Normal Sinus Rhythm, BIDMC Congestive Heart Failure, and Congestive Heart Failure RRI databases. CHF & AF detection with statistical methods and threshold decision making	91.08	88.09	94.06
Liu and Gao 2017 [56]	Physionet NSR and CHF databases. Multiscale Entropy Analysis & SVM	85.6	-	-
Chenet et al. 2017 [57]	Physionet CHF and NSR databases. DL	72.41	-	-

**Table 5 ijerph-17-06313-t005:** Selected work on HR based diabetes detection, since 2016.

Author	Data and Method	Performance		
		Acc/%	Sen/%	Spe/%
Yildirim et al. 2019 [67]	Private data set. DL	97.62%	100%	-
Xiao et al. 2018 [68]	Private data set. Multiscale Cross-Approximate Entropy & Statistical analysis	-	-	-
Swapna et al. 2018 [69]	Private data set. DL	95.7	-	-
Swapna et al. 2018 [70]	Private data set. DL	90.9	-	-
Wilsonet et al. 2017 [71]	Private data set. Linear analysis methods & statistical evaluation	-	-	-

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
