# Peer review of "A Smart Service Platform for Cost Efficient Cardiac Health Monitoring"

_ijerph, 2020, doi:10.3390/ijerph17176313_

Round 1
Reviewer 1 Report
Overall the paper is an interesting paper about the use of Heart Rate data - which can be easily captured through sensors attached to the body - for monitoring a range of diseases.
The paper contains a s significant literature review about the use of this data. It seems there is significant more trials to be done to demonstrate that this approach is clinically acceptable for many of these diseases - including regulatory approval of using HR data - but that is outside the scope of the paper.
Section 3 describes the platform architecture which is a fairly straight forward N-tier level architecture. There are a number of points in the architecture where standards could be considered for capturing and representing the data - such as the DB module which seems to have a free text field "feedback"? Using a structured data approach - using a standard such as FHIR or OpenEHR along with a terminology, such as SNOMED for clinical diagnosis, ensures better data for analysis.
The discussion states that the proposed system is for tackling "public health problems caused by non-communicable diseases". This is where it is unclear where the platform is targeted and how it might be used. Is it envisaged to be a diagnosis platform which is used at a large population level as a screening tool for these diseases? If so, how would the platform be used in practice? i.e. how would the service be prescribed to patients?
The alternatives - as either an investigative tool for new diagnosis or an ongoing monitoring tool for existing patients - are less attractive. Patients with suspected diseases have existing pathways to a quick diagnosis and people with existing diagnoses already have treatment and monitoring pathways. In either of these cases the paper isn't clear on the platform over the existing pathways.
These application problems seem to be the trickiest for such a platform at the moment. If clinical trials show the value of large scale HR monitoring then platforms of this sort may have a potential role in the future.
Author Response
Dear reviewers, thank you very much for the thorough assessment of our document. The points of criticism have helped to sharpen the manuscript focus and address and weakness in the text.
Please allow us to respond to the individual concerns in the text below. The text contains the questions raised by Reviewer 1.
Overall the paper is an interesting paper about the use of Heart Rate data - which can be easily captured through sensors attached to the body - for monitoring a range of diseases.
Answer:
Thank you for the summary.
The paper contains a s significant literature review about the use of this data. It seems there is significant more trials to be done to demonstrate that this approach is clinically acceptable for many of these diseases - including regulatory approval of using HR data - but that is outside the scope of the paper.
Answer:
Indeed this is outside the scope of the paper. The ground for regulatory approval for this type of study is constantly shifting. From a technical perspective, the analysis results are more valuable than the raw data. For example, an attacker who captures the raw data, must do the analysis. Capturing the patient record is more valuable, because this document contains the expert analysis. Apart from the direct use of the data to diagnose disease for an individual, there is also the potential for metaanalysis. For example, with sufficient heart rate data, it might be possible to map, detect, and predict the spread of diseases. This can go far beyond benefiting an individual patient.
Section 3 describes the platform architecture which is a fairly straight forward N-tier level architecture. There are a number of points in the architecture where standards could be considered for capturing and representing the data - such as the DB module which seems to have a free text field "feedback"? Using a structured data approach - using a standard such as FHIR or OpenEHR along with a terminology, such as SNOMED for clinical diagnosis, ensures better data for analysis.
Answer:
Indeed there is a free text field `feedback’ in the user interface. That text field was included to facilitate the need for feedback from the medical expert to the patient. We recognized that need during regular discussions with medical experts.
Using a structured data approach, such as FHIR or OpenEHR together with SNOMED for clinical diagnosis is certainly a way forward. To be specific, if such a backchannel is to be implemented it should use such a structured approach.
Stepping back from the technical aspect of text field `feedback’ and contemplating on the nature of the feedback instead, we recognize a range of issues that need addressing. First and foremost, the main feedback for the patient should be the diagnosis. This result should be communicated via standard channels, i.e. a face to face consultation with the medical expert. Looking forward, there will be situations where there is insufficient time to use standard channels. For example, a service indicates that a myocardial infarction is likely to happen soon. The heart health app might be a good way to facilitate rapid response feedback. Apart from that primary purpose feedback, there is also non-essential feedback. This feedback centers on patient engagement and patient monitoring. For that purpose, a structured data approach might not be optimal. The special use case we had in mind unfolds as follows: the physician support module indicates that there is no heart rate data coming from a specific patient, i.e. patient disengagement. In an initial development phase the required feedback could be facilitated by a nurse encouraging the patient to wear the sensor.
Based on that discussion, we have extended the paper as follows:
“The feedback module can be used to disseminate rapid response warnings when an immediate health risk was detected and it can be used for patient compliance, i.e. a request to wear the sensor. A diagnosis should be communicated via the standard channels established by the healthcare provider.”
The discussion states that the proposed system is for tackling "public health problems caused by non-communicable diseases". This is where it is unclear where the platform is targeted and how it might be used. Is it envisaged to be a diagnosis platform which is used at a large population level as a screening tool for these diseases? If so, how would the platform be used in practice? i.e. how would the service be prescribed to patients?
Answer:
We see the health care provider as the service customer and a commercial entity as the service provider. The medical device technology, which has a similar concept is Holter monitoring. The Holter monitor is used for ECG capturing and specialized software is used to support the cardiologist. A service, based on the proposed service platform works in a similar way. A cardiologist, representing the healthcare provider, initiates the measurement setup, i.e. the patient is registered with the service platform and the heart rate measurement is verified. After that, the patient is free to return home. In the patient environment, the sensor will continue to deliver heart rate data and the deep learning algorithms will be engaged in disease detection. Unlike with Holter monitoring, there is no technical limit on the duration of the measurement process, i.e. standard Holter monitoring lasts for 72h. Once the measurement duration is over and the Holter monitor is returned, specialized software is used to support the cardiologist with data interpretation. The proposed service platform proves that support in real time, hence there is no need for the patient to return the heart rate monitor. The detailed mechanics of running the service depends on national or even regional regulations. For example, in England, health care providers have their own medical cloud servers which could execute the processing part of the service. Thus, a healthcare provider purchases the ability to monitor a specific number of patients.
The alternatives - as either an investigative tool for new diagnosis or an ongoing monitoring tool for existing patients - are less attractive. Patients with suspected diseases have existing pathways to a quick diagnosis and people with existing diagnoses already have treatment and monitoring pathways. In either of these cases the paper isn't clear on the platform over the existing pathways.
Answer:
Initially we want to address the need for individual patients by improving existing diagnosis and treatment monitoring pathways. We envision atrial fibrillation detection as a trailblazer service which could help to establish the necessary infrastructure. Once that infrastructure is established, more services can be added based on the service platform concept. For example, it is possible to monitor heart arrhythmia and at the same time sleep apnea. Due to the large amount of communality, the cost for offering additional services will be small for the service provider. This might lead to an ever increasing number of heart rate measurements. Based on the large amount of measurements, it is possible to track public health problems caused by non-communicable diseases. Opportunities might arise to establish new correlations, and thereby improve risk prediction.
Currently, a major drawback for AI based computer-aided diagnosis is the lack of reliable datasets for training and testing the learning algorithms. Particularly, data taken from patients in an early disease stage are not available. With the proposed service platform, there is no technical limit on the observation duration. Hence, it is possible to capture the onset of disease. Having a large number of measurements implies that there is a large possibility of capturing heart rate signals from a patient in the early disease stage. It might even be possible to document the unfolding of a disease with heart rate data. Such data is valuable for improving the AI algorithms which underpin the services offered by the proposed service platform.
These application problems seem to be the trickiest for such a platform at the moment. If clinical trials show the value of large scale HR monitoring then platforms of this sort may have a potential role in the future.
Answer:
The platform concept is relevant for the service provider, i.e. medical equipment manufacturers. A health care provider gets offered a service for a specific cost. We predict that there will be a `breakthrough’ service which will require heart rate monitoring in the patient environment. For example, there are sound medical and economical reasons to do atrial fibrillation detection based on heart rate signals. These reasons themselves might be sufficiently persuasive to build up the required infrastructure. With our paper, we show that real time heart rate capturing has the potential to seed a wide range of services. To be specific, we show that there is communality for heart rate based disease detection. Hence, many modules can be reused, which keeps the cost for new services down. We predict that new services combined with lower cost will result in more sensors. That move towards big data might provide business as well as healthcare opportunities.
Reviewer 2 Report
The authors describe the value of a smart service platform for cost efficient cardiac health monitoring, which can be used to support the diagnosis of a wide range of diseases including CVD, diabetes and sleep disorders. This is an interesting and topical article. The paper is written well and the conclusions appear to be justified based on the information presented. However, I have a few recommendations for consideration.
- Abstract. Line 1: "Aim" - I am not sure if this is a study, to me it is a review.
- Background. Line 109: Please give USD amount in brackets for the costs indicated in Pounds Sterling.
- Tables: For Tables 1-5, can the authors indicate what the accuracy, specificity as well as sensitivity are based on. Also, it would be helpful to the reader if sample size for each on these cited studies could be provided.
- Discussion. Do the developed algorithms take into account the diagnostic difficulties of CVD in women. Also, what would be the impact/influence of ethnicity on developing machine learning predictive tools.
Author Response
Dear reviewers, thank you very much for the thorough assessment of our document. The points of criticism have helped to sharpen the manuscript focus and address any weakness in the text.
Please allow us to respond to the individual concerns in the text below. The text contains the questions raised by Reviewer 2.
Reviewer #2
The authors describe the value of a smart service platform for cost efficient cardiac health monitoring, which can be used to support the diagnosis of a wide range of diseases including CVD, diabetes and sleep disorders. This is an interesting and topical article. The paper is written well and the conclusions appear to be justified based on the information presented.
Answer:
Thank you for the summary.
However, I have a few recommendations for consideration.
- Line 1: "Aim" - I am not sure if this is a study, to me it is a review.
Answer:
We have changed the study type to review.
- Line 109: Please give USD amount in brackets for the costs indicated in Pounds Sterling.
Answer:
Done
- Tables: For Tables 1-5, can the authors indicate what the accuracy, specificity as well as sensitivity are based on. Also, it would be helpful to the reader if sample size for each on these cited studies could be provided.
Answer:
We have included some details on the data used for the studies.
However, it is almost impossible to establish a fair comparison between individual studies on computer-aided diagnosis. The provided performance measures are meant as a guide for the reader on the possible classification performance. These performance figures were included to support our core premise, namely that heart rate can be used to support the diagnosis for a range of diseases.
- Do the developed algorithms take into account the diagnostic difficulties of CVD in women. Also, what would be the impact/influence of ethnicity on developing machine learning predictive tools.
Answer:
The direct answer to that question must be no, because the datasets do not allow to extract the required knowledge. However, we are aware of the fact that a learning system can only be so good as the data that was used for training. To improve on that situation, continuous learning is required. In terms of system structure, involving the human expert in the decision-making process is of major benefit, because human ingenuity can assist to push the envelope when it comes to increase in the available knowledge. To be specific, the human expert has increased knowledge when a difficult case is resolved. Through continuous learning, that knowledge can be assistive to improve the service platform.
Reviewer 3 Report
Dear authors,
I read your article 'A smart service platform for cost efficient cardiac health monitoring' with interest and find it very clearly disseminated. Despite this clarity, I also find some aspects underpresented or not enough considered.
My main feedback is that the manuscript focusses on the technical side of the platform without considering the human factor, which, in my opinion, is a fundamental part of the platform as it stands now. I will try to clarify this more in the following comments.
Title - Introduction
From the title and the introduction it seems that you developed a platform and tested its use in the different disease you mention (AF, CVD, diabetes, sleep disorders). However, as the paper continues, you give abundant information about the practical use and the ideology behind the platform, but information about the actual testing is very scarce.
Background
This section reads like a review of existing literature. Was this done according to a predefined protocol? Which databases were searched? Is this an overview of all existing literature about deep learning/classical machine learning in AF, CVD,... or is it just a snapshot?
Line 153: "The disease is more prevalent with age,..." Age is just one of the (uncontrollable) risk factors for the development of CVD. Other controllable risk factors for CVD are e.g. high blood pressure, high cholesterol levels, obesity, diabetes mellitus, stress,...
System architecture
I have mixed feelings about the description of the healthcare provider as 'customer' of the platform, while according to my interpretation the healthcare provider is a necessary part of the platform at this moment.
This part of the paper seems like a good place to discuss the protection of sensitive patient data, which is lacking at the moment. It is clear that login profiles are used that are password protected, but is the data that is send to the cloud als encripted, or are there any other measures in place to protect this data?
Line 233 + 259-262: Here you mention very briefly that you tested your platform. A suggestion is that you only describe the different modules without referencing to the testing and then add a separate paragraph combining all the little pieces about the testing.
Line 233 + 237: The reference format is different. You have reference 3 and 4 in superscript and I think these are not the correct references in the reference list?
Discussion
As a healthcare provider, I have some concerns that can be highlighted in the discussion and might create a little bit more of a nuance.
A healthcare provider should make a diagnosis based on the HR measurements of a commercially available HR sensor? Although certain HR sensors have proven to be very valid and reliable, differences exist. Maybe comment on the validity of heart rate sensors or state which valid heart rate sensor is inherently connected to the platform.
If I understand correctly, the healthcare provider is only consulted when an alarm is given. This form of double diagnosis seems very promising in detecting fals positives, but despite the very high accuracy of your deep learning model, fals negatives could be missed.
This brings me to the question of responsibility. You state that the responsibility still rests with the human decision maker (Line 407). If everything is monitored all the time and send to a database accessible by the physician, who is responsible for a missed diagnosis in case the DL fails? This implies that a patient with a missed diagnosis by the platform can e.g. sue the healthcare provider, because the data was there and nothing was done with it.
Although I agree that this platform can facilitate the work of the physician, it is also possible that patients will contact the physician more due to the informal character of the communication. Also this might result in the physician taking on more patients, which results in a maintained burden.
A rather trivial remark, line 325: does the patient acquire the HR sensor? Because if not, it seems to me that the patient still has to return to the hospital to return the HR sensor (still 2 trips). The only added value being that interpretation of the HR signal can already occur during the period that the patient is wearing the sensor and not afterwards.
General conclusions
I am very interested in these kind of topics and cheer on every effort to facilitate disease detection. As you probably noticed, my comments are formulated from a healthcare perspective, rather than a technical perspective. As the healthcare provider is likely to be the end-user of this platform I think it wouldn't do any harm to shine a light on these concerns even if it was only to inspire future research projects.
Kind regards
Author Response
Dear reviewers, thank you very much for the thorough assessment of our document. The points of criticism have helped to sharpen the manuscript focus and address and the weakness in the text. Please allow us to respond to the individual concerns in the text below.
I read your article 'A smart service platform for cost efficient cardiac health monitoring' with interest and find it very clearly disseminated. Despite this clarity, I also find some aspects underpresented or not enough considered.
Answer:
Thank you for the summary.
My main feedback is that the manuscript focuses on the technical side of the platform without considering the human factor, which, in my opinion, is a fundamental part of the platform as it stands now. I will try to clarify this more in the following comments.
Answer:
We have a scientific interest in the platform methodology, because this provides a framework to reason about communality and distinctiveness of healthcare technology. On a technical level, communality constitutes a cost function which we can attempt to optimize in future work. With that, we follow the idea that technology underpins healthcare services.
Despite this technological perspective, we are aware of the delicate interplay between human factor, science, and technology development. These aspects become more relevant when we transit from scientific exercise to practical projects and commercial ventures. Hence, your review has benefitted us.
Title - Introduction
From the title and the introduction it seems that you developed a platform and tested its use in the different disease you mention (AF, CVD, diabetes, sleep disorders). However, as the paper continues, you give abundant information about the practical use and the ideology behind the platform, but information about the actual testing is very scarce.
Answer:
We have a working prototype of the platform, which can be used to understand the human side of the medical tool. The platform is a proposal. After ethical approval, actual testing of the platform idea in a clinical setting requires project funding. To obtain project funding, we have to develop the idea and methodology. The paper constitutes a step towards this goal.
Apart from this direct answer, there is also the principle of extending knowledge by making connections that did not exist or were not clear previously. The authors are subject matter experts for computer aided diagnosis. As such, we are always considering future directions for our research work. This type of paper, and indeed the discussion we are having, helps us to sharpen our argument.
As a direct result of the review process, we have changed the paper type from research paper to review paper.
Background
This section reads like a review of existing literature. Was this done according to a predefined protocol? Which databases were searched? Is this an overview of all existing literature about deep learning/classical machine learning in AF, CVD,... or is it just a snapshot?
Answer:
We have adopted an expert review style for this paper. As such, the background provides a relevant snapshot of knowledge about the diseases, to aid the understanding for readers with a technical background. The background section helps to establish the core prerequisite for the proposed service platform, namely that heart rate can support the diagnosis of many diseases.
Line 153: "The disease is more prevalent with age,..." Age is just one of the (uncontrollable) risk factors for the development of CVD. Other controllable risk factors for CVD are e.g. high blood pressure, high cholesterol levels, obesity, diabetes mellitus, stress,...
Answer:
Excellent point. We have extended the paper as follows:
“CVDs are more prevalent with age, due to prolonged accumulation of plaque and vascular aging [37,38]. Other more controllable risk factors for CVD are e.g. high blood pressure, high cholesterol levels, obesity, diabetes mellitus, and stress [39,40].”
System architecture
I have mixed feelings about the description of the healthcare provider as 'customer' of the platform, while according to my interpretation the healthcare provider is a necessary part of the platform at this moment.
Answer:
Our reasoning was that medical technology offers a service to healthcare providers. Traditionally, medical technology took the form of physical equipment, such as measurement instruments, i.e. MRI and ultrasound scanners. For a medical device manufacturer, the healthcare providers are customers. In this paper, we have adopted the same perspective, i.e. the service platform has the capability to offer the service of computer support for the diagnosis of CVD, diabetes, and epilepsy. It is important that this service comes from an external entity, such as a commercial company, because the knowledge on how to offer this service can benefit multiple healthcare providers.
This part of the paper seems like a good place to discuss the protection of sensitive patient data, which is lacking at the moment. It is clear that login profiles are used that are password protected, but is the data that is send to the cloud as encrypted, or are there any other measures in place to protect this data?
Answer:
Data security is an important point. The service platform requires measurements in the patient environment. These measurements can be conveniently communicated with wireless technology. Unfortunately, the wireless signals are easily captured when an antenna is placed in the vicinity of the patient. As such, wireless technology is an example of a physical problem solution which provides convenience at the price of reduced security. Data encryption is one way of addressing the security problem. Our service platform prototype was based on commercially available HR sensors, namely Polar H10 and Polar H7. As such, these devices use standard Bluetooth encryption to communicate the HR data from the sensor to the mobile phone. From the mobile phone to the cloud server no additional encryption was used. The lack of end-to-end (from sensor to cloud server) encryption and the fact that the HR sensors were not CE marked makes them only suitable to prove the service platform concept but they are not suitable for clinical studies.
Another aspect for the raw data capturing is that a potential attacker faces the same challenge as we do: data interpretation. We propose a hybrid decision-making process where an AI algorithm supports a medical expert during diagnosis. As far as we are aware, this is the only use of individual physiological measurements. More relevant from an individual's risk perspective is the diagnosis, which should be included in the electronic patient record. However, keeping the patient record safe is not a particular problem for the service platform.
To reflect that discussion, we have extended the future work section of the paper as follows:
“... In order to involve patients and medical experts in the next validation step, we have to address shortcomings in the way the sensor data is handled. To be specific, currently only standard protocols were used for the Data acquisition module. These protocols follow no or only weak data security standards. Hence, the data is at risk of being captured with negative consequences for the patient. This needs to be addressed with end-to-end encryption from the sensor to the cloud server. Furthermore, for clinical studies involving patients, the sensor and data handling technology should conform to health, safety, and environmental protection standards. The sensors used to test the platform prototype fall short of the health protection standard required for medical devices.”
Line 233 + 259-262: Here you mention very briefly that you tested your platform. A suggestion is that you only describe the different modules without referencing to the testing and then add a separate paragraph combining all the little pieces about the testing.
Answer:
We created and tested a functional prototype. To document this activity, we extended the paper as follows:
“3.8. Testing
The HHMSP was tested with the AF monitoring service. By doing that, we tested all common modules for data handling and the specific modules which establish the diagnosis support. To verify all aspects of the AF monitoring service, we registered two healthy volunteers with the DB module.Subsequently the volunteers were fitted with HR sensors and their data was communicated and processed by the HHMSP. To be specific, their data was processed with the deep learning module which executed the AF specific DL model. However, the deep learning results were not verified by a cardiologist. Therefore, no diagnosis was established and the results have only an indicative character.
To validate both deep learning and physician support modules for the AF monitoring service,we streamed benchmark data from Physionet’s MIT-BIH Atrial Fibrillation Database [74] to the data acquisition module. Figure 4 shows a screenshot of the physician support module which details both deep learning and feature extraction results for the benchmark data. Apart from visual inspection,enabled by the physician support module, we also compared the deep learning module results for the benchmark data with the results for the same benchmark data obtained by a previous study on the automated detection of atrial fibrillation using long short-term memory network with RR interval signals [15]. In that study, we used the same model as in the deep learning module. We established that the deep learning results were the same. /with that, we could validate both Data acquisition and Deep learning modules. Furthermore, that validation confirmed that the AF monitoring service achieved 99.77% accuracy with benchmark data.”
Line 233 + 237: The reference format is different. You have reference 3 and 4 in superscript and I think these are not the correct references in the reference list?
Answer:
In line 233 and 237 the superscript 3 and 4 refers to footnotes at the bottom of the page. In general, we have used footnotes for web pages. Unfortunately, the typesetting program did not produce hyperlinks where you click on the footnote reference (superscript number) and you get linked to the corresponding footnote.
Discussion
As a healthcare provider, I have some concerns that can be highlighted in the discussion and might create a little bit more of a nuance.
Answer:
Thank you for the summary.
A healthcare provider should make a diagnosis based on the HR measurements of a commercially available HR sensor? Although certain HR sensors have proven to be very valid and reliable, differences exist. Maybe comment on the validity of heart rate sensors or state which valid heart rate sensor is inherently connected to the platform.
Answer:
Creating that platform prototype was an important exercise, because it showed us the limitation in our understanding and indeed it showed the limitation of readily available technology. The sensors used to verify the platform are an example for technological limitations. They were readily available, but they lack data protection and they are not medically certified. For the platform testing, this is not an issue, because the tests were not done with patient data. However, to move forward towards the creation of a physical problem solution the sensor shortcomings must be addressed. For future clinical studies we plan to use different heart rate sensors which are medically certified.
If I understand correctly, the healthcare provider is only consulted when an alarm is given. This form of double diagnosis seems very promising in detecting fals positives, but despite the very high accuracy of your deep learning model, fals negatives could be missed.
Answer:
Indeed this is a shortcoming, however this is a shortcoming for all decision-making systems - including human expert decision-making. One of the main benefits of the proposed service platform is that there are no technical limitations for the observation duration. Hence, it is possible to capture the heart rate during acute episodes of the disease. This is particularly important for atrial fibrillation, because paroxysmal atrial fibrillation might be asymptomatic for a long time. The deep learning algorithm might misclassify AF beats as being normal; however the performance test results, accuracy, sensitivity, and specificity, indicate that it is unlikely to miss a complete atrial fibrillation episode.
Current disease detection methods suffer from the false negative problem as well. However, in contrast to the proposed service platform, a negative diagnosis will stop the measurement. For example, atrial fibrillation is detected based based on 72 h ECG from Holter monitors. If the diagnosis is false negative, the measurement will not be continued, at least not automatically. In contrast, the service platform would continue to monitor and therefore it might detect the next symptomatic episode.
Furthermore, for atrial fibrillation, false positive is the more dangerous case, because the treatment carries the risk of bleeding, which can lead to death.
This brings me to the question of responsibility. You state that the responsibility still rests with the human decision maker (Line 407). If everything is monitored all the time and send to a database accessible by the physician, who is responsible for a missed diagnosis in case the DL fails? This implies that a patient with a missed diagnosis by the platform can e.g. sue the healthcare provider, because the data was there and nothing was done with it.
Answer:
The proposed service platform is an adjunct tool for clinicians to confirm the diagnosis and thereby reduce human error. To sharpen the argument, we need to differentiate between two different types of failure. The first type of failure is an engineering problem that happens in a well-defined space. In its simplest form, it can be the
case of someone who did not plug a cable in, and the system failed to detect the missing link. This is a risk which exists for all medical devices. If such a problem happens, there might be a case for suing the service provider.
For the second type of failure, we have to recognise that the proposed service platform is based on deep learning algorithms and this algorithm operates outside a well-defined engineering space. The immense state space makes it impossible to trace into a deep learning model with the intent to determine what caused the wrong decision. The goal of the algorithm is to apply knowledge extracted from an incomplete learning dataset to a practical case, i.e. patient data. Applying the knowledge might fail because the technology is not sufficiently advanced, or it might be impossible to extract the needed information from the training dataset. Both cases result from a lack of knowledge which is inherent in all science and technology. In other words, it is impossible to use tomorrow's technology today. Based on this argument there is little scope to mount a successful court case.
The argument that nothing was done with the data is not valid, when we assume that that deep learning algorithm analyzed the data - second failure type. The medical device failed because of insufficient information or knowledge. This is regrettable, but it would be even more regrettable not to learn from the mistake. To be specific, each failure constitutes new knowledge that can be used to improve the proposed service platform.
Although I agree that this platform can facilitate the work of the physician, it is also possible that patients will contact the physician more due to the informal character of the communication. Also this might result in the physician taking on more patients, which results in a maintained burden.
Answer:
The feedback channel goes from the clinician to the patient. That feedback channel is used only if there is a problem. Hence the patient will not be disturbed by the clinicians as long as there is no problem. This type of system will have more value where the physicians are not readily available for the patients.
A rather trivial remark, line 325: does the patient acquire the HR sensor? Because if not, it seems to me that the patient still has to return to the hospital to return the HR sensor (still 2 trips). The only added value being that interpretation of the HR signal can already occur during the period that the patient is wearing the sensor and not afterwards.
Answer:
This is a question for the business model which is shaped by commercial interests and healthcare regulations. The closest well established medical technology to the proposed service platform is Holter monitoring. The patient does not acquire the Holter monitor and the equipment has to be returned upon completion of the measurement.
The main benefit of the proposed service platform is the fact that there are no technological limits for the observation duration. Hence, there is no technological need to return the measurement equipment. There might be a human need to change the data acquisition arrangements. But this is a significant improvement, because the service platform adopts to patient needs and not the other way around.
General conclusions
I am very interested in these kind of topics and cheer on every effort to facilitate disease detection. As you probably noticed, my comments are formulated from a healthcare perspective, rather than a technical perspective. As the healthcare provider is likely to be the end-user of this platform I think it wouldn't do any harm to shine a light on these concerns even if it was only to inspire future research projects.
Answer:
Physiological signals are key for big data medicine. For the service platform we stayed in the area of computer aided diagnosis, because we have clear evidence that such a system can improve the current state of health care. However, the promise of big data goes further. Once we have a sufficient amount of sensor information, new application areas arise. There are analogs to smartphones, which were initially only used as sophisticated communication devices. By now, tracking the phones on the street gives rise to predict the traffic situation. Similarly, having physiological signals from a large number of patients, might help to predict the spread of diseases beyond CVD. Like with all new technology, ethical issues exist, but many of them come from the political domain which is constantly shifting.
In the paper, we put forward that the healthcare providers should play an important role in big data medicine. However, technology moves fast in this area and measurement as well as processing becomes less expensive over time. In the future, technology companies might offer potent disease detection as a gadget, where medical experts are excluded from the decision-making process. Similar technology, like e-mail (cloud) and voice assistant (AI), is free. Therefore, technology companies might be able to offer these services for free as well.
Round 2
Reviewer 3 Report
Dear authors,
Thank you very much for your elaborate answers to my questions. They helped me to better understand the goals of this paper.
I am satisfied with the changes you made to the manuscript and the clarifications you gave.
I look forward to reading about the next steps in this research line.
Kind regards and good luck!